# Optimized Adaptive Radiotherapy with Individualized Plan Library for Muscle-Invasive Bladder Cancer Using Internal Target Volume Generation

**DOI:** 10.3390/cancers14194674

**Published:** 2022-09-26

**Authors:** Yoon Young Jo, Ji Woon Yea, Jaehyeon Park, Se An Oh, Jae Won Park

**Affiliations:** Department of Radiation Oncology, Yeungnam University Medical Centre, University of Yeungnam College of Medicine, Daegu 42415, Korea

**Keywords:** adaptive radiotherapy, plan library, survival, toxicity, urinary bladder neoplasms

## Abstract

**Simple Summary:**

The bladder is a mobile target and is subject to filling variation. This poses a considerable challenge for effective radiotherapy (RT) delivery. We applied an internal target volume to the plan library to resolve intra-fractional errors caused by bladder filling during treatment. Adaptive radiotherapy using ITV is easy to perform and a feasible treatment approach. In this study, image-guided RT-based adaptive RT showed good survival outcomes with a high local control rate.

**Abstract:**

The bladder is subject to filling variation, which poses a challenge to radiotherapy (RT) delivery. We aimed to assess feasibility and clinical outcomes in patients with bladder cancer treated with adaptive RT (ART) using individualized plan libraries. We retrospectively analyzed 19 patients who underwent RT for muscle-invasive bladder cancer (MIBC) in 2015–2021. Four planning computed tomography (CT) scans were acquired at 15-min intervals, and a library of three intensity-modulated RT plans were generated using internal target volumes (ITVs). A post-treatment cone-beam CT (CBCT) scan was acquired daily to assess intra-fraction filling and coverage. All patients completed the treatment, with 408 post-treatment CBCT scans. The bladder was out of the planning target volume (PTV) range in 12 scans. The volumes of the evaluated PTV plans were significantly smaller than those of conventional PTV. The 1-year and 2-year overall survival rates were 88.2% and 63.7%, respectively. Of eight cases that experienced recurrence, only two developed MIBC. There were no grade 3 or higher RT-related adverse events. ART using plan libraries and ITVs demonstrated good survival outcomes with a high local control rate. Irradiated normal tissue volume and treatment margins may be reduced through this approach, potentially resulting in lower toxicity rates.

## 1. Introduction

Bladder cancer is the 10th most commonly diagnosed cancer worldwide, with approximately 573,000 new cases and 213,000 deaths occurring annually. It is more common in men than in women and is the sixth most commonly diagnosed cancer in men worldwide [1]. In conjunction with radical cystectomy, definitive radiation therapy (RT) is a well-established alternative treatment for localized muscle-invasive bladder cancer (MIBC), forming part of a multimodal treatment. Trimodal therapy (TMT), consisting of maximal transurethral resection of the bladder tumor (TURBT) followed by RT with concurrent chemotherapy, offers the advantage of bladder preservation and has demonstrated good survival outcomes comparable to those of radical cystectomy in a well-selected MIBC population [2,3,4,5,6,7].

RT for bladder cancer is technically challenging because of significant variations in bladder volume, shape, and position. Variations in bladder volume cause large inter/intra fractional errors. The conventional technique, using large 1.5–3 cm margins around the bladder to involve a sufficient clinical target volume (CTV), results in a remarkable level of irradiation to organs at risk (OARs). This increases acute toxicity, causes treatment interruptions, and increases late toxicity [8].

Adaptive radiotherapy (ART) was introduced to circumvent this challenge and is becoming more widely available in clinical practice [9,10,11,12,13]. The plan-of-the-day (POD) approach for ART utilizes daily imaging to select the best-fit plan from a library of plans. This methodology is shown to be feasible and is generally well-tolerated [14,15]. However, previous evaluations of ART using plan libraries did not reflect bladder changes during treatment. Lalondrelle et al [15]. reported an average bladder volume increase of 9 cm³ (2–3 mm of bladder wall displacement). Caroline et al. [16] reported a median bladder volume increase of 11 cc in 10 min during RT. Therefore, our institution is implementing the POD approach using internal target volumes (ITVs) to solve the problem of bladder filling during treatment.

This study aimed to assess feasibility and clinical outcomes in patients with bladder cancer who were treated with ART using an individualized plan library and ITV generation.

## 2. Materials and Methods

### 2.1. Patient Selection

A retrospective analysis was performed on the data of patients who underwent RT for MIBC (December 2015 to December 2021). Patients with histologically proven transitional cell carcinoma, a primary tumor stage of T2–T4 (American Joint Committee on Cancer TNM staging, 8th edition), and those without distant metastasis were enrolled.

### 2.2. General RT Schedule

Patients received conventionally fractionated RT divided into two components. Elective RT included RT of the whole bladder without elective lymph node irradiation. Boost RT included only the tumor or tumor bed. The radiation schedule was delivered as 44 Gy (22 fractions) to the whole bladder and 20 Gy (10 fractions) to the tumor bed or gross tumor. Whole-bladder irradiation was performed in an empty bladder condition; boost irradiation was performed in a full bladder condition. When boost irradiation was administered after whole-bladder irradiation, urinary frequency and urgency increased, making bladder filling difficult. Therefore, boost irradiation was performed first after the first three patients.

### 2.3. Plan Library Generation

Patient preparation involved ensuring an empty bladder condition, i.e., voiding immediately before undergoing a CT scan. Four CT scans were taken every 15 min (immediately after voiding (CT0) and after 15 min (CT15), 30 min (CT30), and 45 min (CT45)). All patients were scanned in the supine/arm-chest position.

The gross tumor volume (GTV) was defined as the visible tumor or tumor bed. The clinical target volume (CTV) was contoured to encompass the GTV, the whole bladder, and any extravesical spread. In patients with direct invasion into the prostate, the whole prostate was included in the CTV. 

After delineating CTV0 on CT0, then CTV15, CTV30, and CTV45 were delineated using each registered CT15, CT30, and CT45 reference. CTV0 and CTV15 were then combined to form the ITV0–15. ITV15–30 (CTV15 + CTV30) and ITV30–45 (CTV30 + CTV45) were generated in the same way. For the generation of a library of plans, three planning target volumes (PTVs: PTV0–15, PTV15–30, and PTV30–45) were created with 5–7 mm expansion in all directions from the ITV. The study workflow is provided in Figure 1.

### 2.4. Treatment Planning

Radiotherapy planning was used to generate the PTV0–15, PTV15–30, and PTV30–45, respectively. To minimize bladder filling during treatment, it was necessary to shorten the treatment time. Therefore, the treatment plan was decided using dual arc volumetric modulated arc therapy (VMAT). The treatment was performed using Novalis Tx (Varian Medical Systems, Palo Alto, CA, USA); 6 MV or 15 MV was selected according to the patient′s body contour. Dose constraints are listed in Appendix A.

### 2.5. Online Adaptive Radiotherapy Workflow

Patients were asked to void their bladder before treatment. Pre-treatment CBCT scans were acquired and co-registered with the treatment planning CT reference image (CT0), mostly based on bony structures; setup errors were corrected accordingly. The bladder volume in the CBCT images was compared to three plan libraries by a radiation oncologist. The most appropriate PTV that encompassed the bladder was then chosen. The bladder volume was compared to CTV0, CTV15, and CTV30, and the smallest CTV that was appropriate was selected. For example, if the bladder was included in CTV0, then PTV0–15 was selected; if the bladder was larger than CTV0 and included within CTV15, PTV15–30 was selected.

### 2.6. Validation Workflow

The post-treatment CBCT scan was acquired immediately after treatment. A radiation oncologist assessed post-CBCT findings to confirm whether bladder filling was included in the selected PTV during treatment; the time between pre-and post-treatment CBCT scans was used as a surrogate for the overall treatment time. The normal bladder was contoured by another radiation oncologist on each CBCT scan to ensure that target coverage for treatment was possible with the pre-selected library of plans. We performed calculations to confirm the detailed figures for volumes outside the PTV.

After that, the ratio of the selected PTV to the volume obtained by subtracting the selected PTV from the drawn PTV in the post-treatment CBCT was calculated, with the following classifications: “good” delineated a bladder volume that was sufficiently included in the selected PTV; “fair” delineated a bladder volume that was outside the PTV by ≤5%; and “poor” delineated a bladder volume that was outside the PTV by >5% (Appendix A). In addition, a virtual PTV containing the empty bladder expanded with a 1 cm and 1.5 cm margin was contoured to represent conventional non-adaptive treatment plans. By comparing the virtual PTV volume with each plan library volume, we tried to demonstrate that a small PTV volume could reduce the incidence of acute toxicity by reducing the range of irradiation to the surrounding normal organs.

### 2.7. Survival and Toxicity Assessment

Overall survival (OS), progression-free survival (PFS), local recurrence-free survival (LRFS), and distant metastasis-free survival (DMFS) were evaluated. Local recurrence was further evaluated by dividing recurrence into non-muscle-invasive local recurrence (NMILR) and muscle-invasive local recurrence (MILR). OS, PFS, LRFS, and DMFS durations were calculated from the date of surgical resection to the date of recurrence, death, or the follow-up. During treatment and follow-up, the toxicity was assessed and graded according to the Common Terminology Criteria for Adverse Events (version 5.0). The patients were followed up at 3-month intervals for 12 months, at 6-month intervals for ≤3 years, and annually thereafter.

### 2.8. Statistical Analysis

Chi-square tests and Student’s *t*-tests were used to analyze the distributions of categorical and continuous variables, respectively. The Kaplan–Meier method was used to determine survival rates. Log-rank tests and Cox proportional hazards regression models, respectively, were used for univariate and multivariate analyses to determine prognostic risk factors. A *p*-value of <0.05 was considered statistically significant. SPSS statistical software (v.22.0; IBM Corporation, Armonk, NY, USA) was used to perform the statistical analyses.

### 2.9. Ethical Statement

This study was approved by the Institutional Review Board of the Yeungnam University Medical Center (YUMC 2022-08-007) and was conducted in accordance with the principles of the Declaration of Helsinki. The requirement of obtaining informed consent was waived due to the retrospective nature of this study.

## 3. Result

### 3.1. Patient Characteristics

A total of 19 patients were enrolled (December 2015 to December 2021). The median age of the enrolled patients was 80 (range, 66–91) years, and 68.4% were male. Of the 19 patients, 17 had at least one comorbidity. Among these 17 patients, 11 had multiple comorbidities, which made them unsuitable for radical surgery. The patient medical and demographic characteristics are presented in Table 1.

### 3.2. Plan Selection

A total of 826 CBCT scans were evaluated (418 before RT and 408 after RT). Before treatment, the best-fit PTV was chosen by the attending radiation oncologist. The treatment was delivered using PTV0–15, PTV15–30, and PTV30–45 plans for 56.9, 26.5, and 16.6% of the cases, respectively. Individual plan selections are presented in Figure 2.

### 3.3. Comparing ART PTV with Conventional Pseudo-PTV

The mean time between the pre- and post-CBCT scans was 9.8 (range, 7.6–12.2) min. The mean (± standard deviation) volumes for the PTV0–15, PTV15–30, and PTV30–45 plans were 222.3 ± 73.6, 260.4 ± 82.1, and 306.8 ± 98.5, respectively, which were significantly smaller than the non-adaptive conventional PTV volumes (CTV+1 cm, 304.8 ± 101.6, and CTV+1.5 cm, 446.7 ± 144.7) (Figure 3).

### 3.4. Post-Treatment CBCT Bladder Coverage

For each treatment fraction, post-treatment CBCT was co-registered with a planning CT scan. For instance, we evaluated whether the bladder was expanded beyond the daily selected PTV at the end of treatment (Appendix A). A total of 408 post-CBCT scans were evaluated. In 97% of cases, treatment was continued without missing components. The total number of CBCT libraries and each calculation are shown in Appendix A. Of the 408 post-treatment CBCT scans, 12 (0.03%) were classified as “poor” and showed that the bladder was partially outside of the PTV margin. This occurred once in three patients, twice in one patient, three times in one patient, and four times in one patient. In most patients, rapid bladder filling was thought to be a consequence of pre-chemotherapy hydration.

### 3.5. Treatment Outcomes

In terms of feasibility, all 19 patients completed the treatment. The median follow-up time was 16.3 (range, 6.2–77.9) months. The 1-year and 2-year OS rates were 88.2% and 63.7%, respectively. The PFS and LRFS rates at 1 year were 74.5% and 87.5, respectively. Of the six patients who experienced local recurrence, three patients developed non-muscle-invasive bladder cancer and were managed using TURBT. The 1-year and 2-year DMFS values were 87.5% and 76.6%, respectively. Among eight (42.1%) deaths, two (25.0%) were due to disease. The cancer-specific survival (CSS) at 2 years was 100%. All survival graphs are shown in Figure 4 and patterns of disease recurrence are shown in Appendix A. No significant prognostic factors for 2-year OS were identified on univariate or multivariate analysis (Appendix A).

### 3.6. Treatment-Related RT Toxicity

The incidence of acute and late gastrointestinal/genitourinary adverse events after radiotherapy is shown in Table 2. Grade 2 early and late urinary toxicities were seen in one patient each, who presented with gross hematuria. Severe grade 3 or greater acute or late toxicities were not observed.

## 4. Discussion

This study aimed to show the efficacy and safety of adaptive VMAT for bladder preservation in routine clinical practice. The OS and CSS rates at 5 years were 36.4% and 66.7%, respectively. The non-muscle invasive local recurrence survival rates at 1 and 2 years were 100% and 66.7%, respectively (Appendix A), showing a high local control rate. Published long-term outcomes following TMT vary greatly, ranging from 26 to 73% for 5-year OS and 30 to 82% for 5-year CSS [3,17,18]. The rate of survival in our study was comparable, even though only 6 of our 19 patients received induction chemotherapy and only approximately half the patients received concurrent chemo-radiation therapy (CCRT).

Over the years, definitive radiotherapy for MIBC has evolved from a palliative treatment in patients unfit for radical cystectomy to an alternative option for bladder preservation. A review of recent evidence shows results for modern RT techniques that are comparable with those of surgical treatment. A systematic review of 13,396 patients treated in various clinical trials favored TMT over radical cystectomy in regard to 5-year OS (57% versus 52%, *p* = 0.04) [19].

Traditionally, in bladder cancer radiotherapy, an indwelling urinary catheter method (with the instillation of a specific volume of normal saline) is used to reduce variability in bladder size. This method presents the disadvantages of infection risk and irritation during a long course of radiation treatment. With emerging advances in image-guided radiotherapy (IGRT) and intensity-modulated radiotherapy (IMRT), the concept of adaptive radiotherapy was formalized by Yan et al. [20]. Two methodological approaches have been suggested in an attempt to improve the ART planning process. One approach is to use the initial CBCTs acquired in the first week of treatment, and the other is to use multiple pre-treatment CT scans. The concept of a ‘plan of the day’ was initially described by Burridge et al. [21]. These researchers used three PTVs around the bladder, with a variable superior margin. Since then, several dosimetry studies have reported improved target coverage and normal tissue sparing with the use of this approach [15,22,23,24]. However, most previous ART studies did not reflect bladder changes due to urine filling during intra-treatment.

Therefore, we created three PTV libraries by adding ITV references (in consideration of intra-fractional filling) to the repeat-planning CT. Appropriate plan selection (from a choice of three plans) was evaluated on the post-treatment CBCT scans. A total of 408 post-CBCT scans were evaluated. In 97% of cases, treatment proceeded without missing components. In the case of a patient who was classified as “poor” as a result of the calculation, rapid bladder filling was thought to be a consequence of pre-chemotherapy hydration. We note that Turner et al. [25] demonstrated considerable inter-fractional variability of CT-measured bladder dimensions in 20 patients, with a median area of change of 11.1 cm^2^ over the course of treatment. As treatment progressed, residual urine increased due to an increase in bladder vulnerability. Nevertheless, the selection of the plan library in this study was appropriate. In more than half the cases, whole-bladder coverage was sufficient with a small PTV plan (i.e., PTV0–15 was sufficient in 56.9% of the cases).

It has been reported that the use of a plan library results in increased treatment time. The IMRT plan used for treatment delivery is reported to take approximately 21 min because it requires an average of 5–12 min of additional time [26]. In our case, despite the imaging and plan selection process, delivery was possible within a reasonable time frame using the VMAT plan, with a mean of 9 min between the pre- and post-CBCTs. In addition, the maximal treatment time was 12.2 min, such that CTV simulation using 15-min intervals to construct a CT dataset and generate ITVs was sufficient in terms of timing and efficacy. These considerations are important for both the efficiency of allocating radiotherapy resources (i.e., time slots) and minimizing the impact of intra-fractional change on plan selection.

In this study, ART was safely performed for MIBC, with limited toxicity. All patients completed the radiotherapy course, and none experienced severe adverse events (grade ≥3). The mean volume of the PTV libraries was significantly smaller than that for the non-adaptive conventional PTV volumes (CTV+1 cm and CTV+1.5 cm). We note that a dosimetry study evaluating IGRT, conducted by Foroudi et al. [23], examined patients who were treated with daily adaptive IGRT using tomography, showing substantially reduced normal tissue doses (at V45 Gy and V5 Gy). Tuomikoski et al. [27] reported considerable dose reduction in regard to the intestinal cavity (IC) volume. The average IC volume reduction at 45 Gy was 155 cm³. By applying our results to these previous dosimetry studies, it can be inferred that a smaller PTV volume reduced the volume of irradiated normal organs. As a result, it can be inferred that less RT-related toxicity occurred.

The latest form of ART involves online re-planning. Due to advancements in treatment planning algorithms and the development of a magnetic resonance imaging (MRI) linear accelerator, online re-planning may now be a feasible option. Although most MRI-guided ART studies have focused on prostate [28] and gynecological cancer [29], Vestergaard et al [30] compared bladder target coverage between online MRI-guided re-optimization and a plan library approach. These researchers found that MRI-guided ART strategies resulted in a reduction in averaged PTVs as compared to the use of a plan library. However, carrying out online re-planning requires an additional 10 min, and target under-dosing problems can result due to bladder shift. Therefore, additional research is needed prior to implementing this methodology in actual clinical practice.

The limitations of this study include selection bias and a lack of comparative analysis as regards the irradiated normal tissue. Moreover, most of the included patients had numerous comorbidities and were unfit for radical cystectomy. Therefore, when MIBC recurred after treatment, salvage cystectomy could not be performed. Second, the number of enrolled patients was small as the data were collected from a single institution. Consequently, subgroup analyses do not allow for a definitive interpretation of the impact of induction chemotherapy or concurrent chemotherapy. Third, the study design was retrospective. A large-scale prospective observational study is necessary to fully address the abovementioned issues and to draw causal inferences.

## 5. Conclusions

Plan library ART-based bladder preservation using ITV is easy to perform and represents a feasible treatment approach. As demonstrated in this study, IGRT-based adaptive RT, which involves a daily choice from among the generated pre-planned libraries, showed good survival outcomes with a high local control rate. Although more research is required, this method may represent a safe and effective alternative to the use of population-based large PTV margins and may allow for a greater degree of sparing of the surrounding normal tissue. If confirmed, the applied findings of this research may result in improved treatment efficacy with lower toxicity rates.

## Figures and Tables

**Figure 1 cancers-14-04674-f001:**
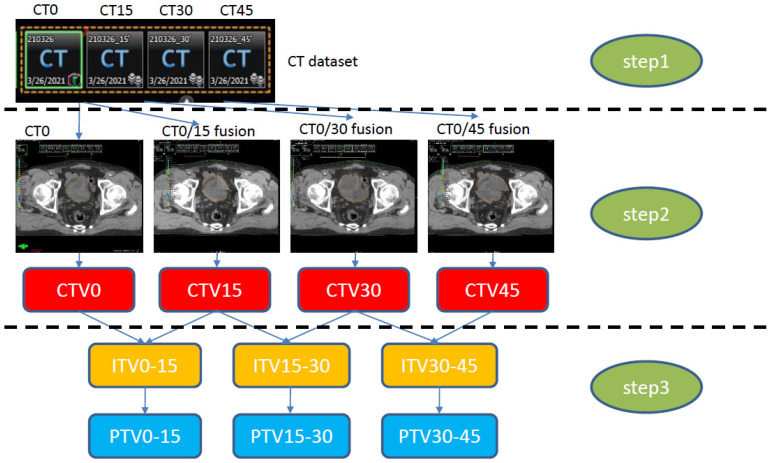
Study workflow. Step 1: CT simulation with an empty bladder (CT0). Sequential CT simulation at 15 min intervals (CT15, CT30, CT45) to construct the CT dataset. Step 2: Four CTV generations (CT0). CTV15 was constructed through fusion of CT0/15 delineated on CT0; the rest of the CTVs were constructed in the same way. Step 3: ITV generation on CT0, constructed by merging each CTV (e.g., ITV0–15: CTV0 + CTV15), with an added 5 mm expansion to generate the PTV. CT, computed tomography; CTV, clinical target volume; ITV, internal target volume; PTV, planning target volume.

**Figure 2 cancers-14-04674-f002:**
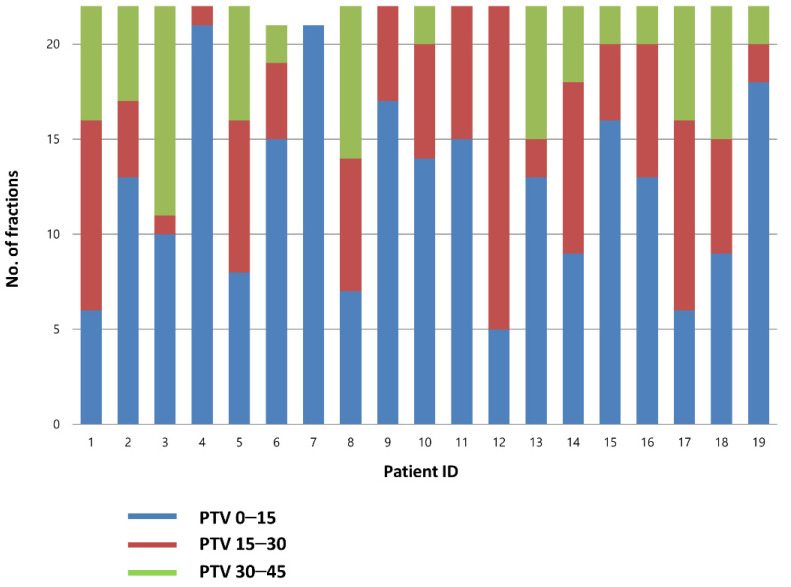
Choice of the PTV in each fraction. PTV, planning target volume.

**Figure 3 cancers-14-04674-f003:**
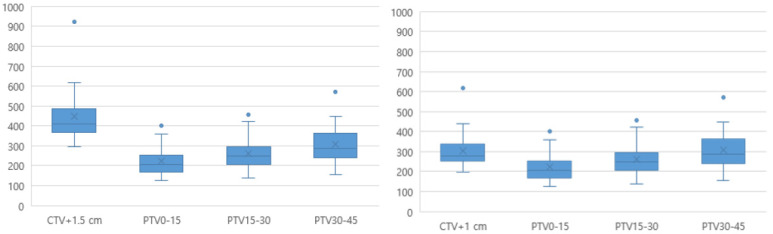
Comparison of the ART-PTV with the conventional PTV. ART, adaptive radiotherapy; CTV, clinical target volume; PTV, planning target volume.

**Figure 4 cancers-14-04674-f004:**
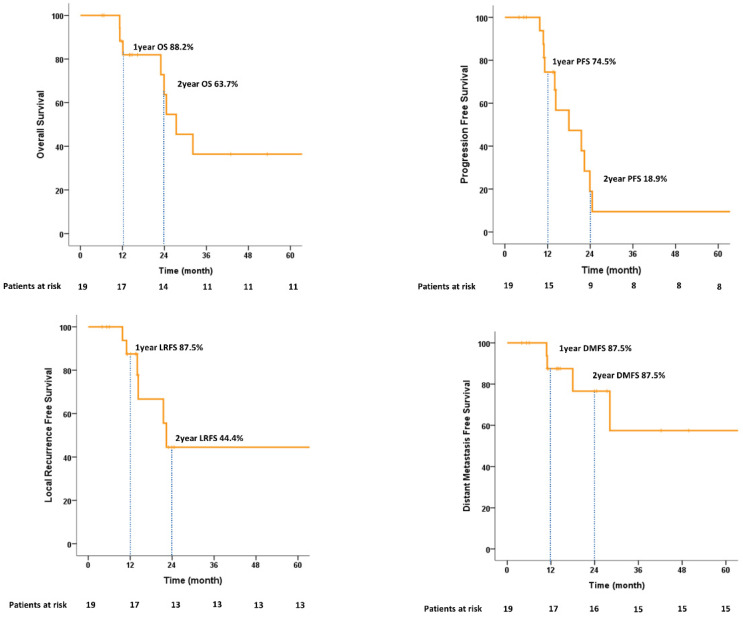
Kaplan–Meier estimates of overall survival, progression-free survival, local recurrence-free survival, and distant metastasis-free survival.

**Table 1 cancers-14-04674-t001:** Baseline characteristics of the patients and tumor.

Characteristics	*n* = 19	
Age, years (median, (range))	80	(66–91)
Sex (*n* (%))		
Male	13	68.4
Female	6	31.6
Charlson Comorbidity Index (median, (range))	3	(2–6)
ECOG status (*n* (%))		
0–1	18	94.7
2–3	1	5.3
T stage (*n* (%))		
T1	0	0
T2	13	68.4
T3	5	26.3
T4	1	5.3
AJCC 8th Stage (*n* (%))		
I	0	0
II	13	68.4
III	6	31.6
IV	0	0
Induction CTx (*n* (%))		
Received	6	31.6
Not received	13	68.4
CCRT (*n* (%))		
Received	10	52.6
Not received	9	47.4
RT dose, Gy (median, (range))	64.0	(62.0–66.0)
RT technique (*n* (%))		
VMAT	19	100
PTV volume, cm³ (mean, ±standard deviation)		
PTV0–15	222.3	73.6
PTV15–30	260.4	82.1
PTV30–45	306.8	98.5
CTV+1 cm	304.8	101.6
CTV+1.5 cm	446.7	144.7
F/U duration, month (median, (range))	14.2	(4.0–76.0)

ECOG, Eastern Cooperative Oncology Group; AJCC, American joint committee of cancer; RT, radiation therapy; CT, chemotherapy; CCRT, concurrent chemoradiation therapy; CTV, clinical target volume; PTV, planned target volume; F/U, follow up.

**Table 2 cancers-14-04674-t002:** RT-related acute and late toxicity.

Toxicity	*n* = 19
Early GI toxicity	
Grade 1	6
Grade 2	0
Grade 3	0
Early GU toxicity	
Grade 1	18
Grade 2	1
Grade 3	0
Late GI toxicity	
Grade 1	3
Grade 2	0
Grade 3	0
Late GU toxicity	
Grade 1	12
Grade 2	1
Grade 3	0
Toxicity	*n* = 19

Toxicity was assessed and graded according to the common terminology criteria for adverse events (version 5.0).

## Data Availability

Data are available upon request from the authors.

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
