# Peer review of "Optimized Adaptive Radiotherapy with Individualized Plan Library for Muscle-Invasive Bladder Cancer Using Internal Target Volume Generation"

_cancers, 2022, doi:10.3390/cancers14194674_

Round 1
Reviewer 1 Report
This research article demonstrates the feasibility of Adaptive radiotherapy for muscle-invasive bladder cancer. In this work, the authors competently address the issue of implementing individualized plan-of-the-day approach to solve intra-fractional variations in bladder filling.
In general, the manuscript is well-written and well-structured, taking into account that Adaptive radiotherapy (ART) tends to replace conventional RT in several occasions.
The work is therefore publishable after some additional considerations listed below.
Line 78. The authors mention the acquisition of a plan library with an empty bladder condition. What is the method of practice for the boost RT?
Line 183. At the study, virtual PTVs of 1 cm and 1.5 cm were also contoured. As it can be seen at Table 1, the margin of 1 cm corresponds to PTV volume of 304.8 cc which is the same with the largest plan library volume of PTV30-45 (306.8 cc). Since the margins adopted for ART PTVs are already 5-7 mm (Line 90), it is questionable whether a plan library generation is necessary, taking into consideration the extra workload time and effort, the extra treatment time, as well as the double CBCT patient exposure. The authors should point out the need for their suggested ART methodology. Probably, this could be more essential in stereotactic treatments, where the delivered dose is much higher.
Line 193. Please refer that these 12 post-RT CBCTs belong to the “poor” classification group.
Line 221. In this paragraph, the authors compare the 2-years OS and CSS from their study with the 5-years outcomes from other studies. It would be preferable if the authors compared the outcomes of the same time periods, in order to critically evaluate the results from the Adaptive RT.
Additionally, the adverse events (acute and late) from adaptive and conventional RT should be compared.
Line 281. The authors mention dosimetric results from other studies. In my opinion, this study should also present dosimetric results for the organs at risk, compare them with the conventional RT doses, and correlate them with the statement of ‘less RT-related toxicity’.
Author Response
Reviewer #1
Dear authors,
This research article demonstrates the feasibility of Adaptive radiotherapy for muscle-invasive bladder cancer. In this work, the authors competently address the issue of implementing individualized plan-of-the-day approach to solve intra-fractional variations in bladder filling.
In general, the manuscript is well-written and well-structured, taking into account that Adaptive radiotherapy (ART) tends to replace conventional RT in several occasions.
The work is therefore publishable after some additional considerations listed below.
Response: We sincerely appreciate your positive feedback.
- Line 78. The authors mention the acquisition of a plan library with an empty bladder condition. What is the method of practice for the boost RT?
Response: Boost irradiation was performed in full bladder condition. This is mentioned in the Materials and Methods, section “2.2 General RT schedule” (Line 72–73), as follows:
“Whole-bladder irradiation was performed in an empty bladder condition; boost irradiation was performed in a full bladder condition.”
- Line 183. At the study, virtual PTVs of 1 cm and 1.5 cm were also contoured. As it can be seen at Table 1, the margin of 1 cm corresponds to PTV volume of 304.8 cc which is the same with the largest plan library volume of PTV30-45 (306.8 cc). Since the margins adopted for ART PTVs are already 5-7 mm (Line 90), it is questionable whether a plan library generation is necessary, taking into consideration the extra workload time and effort, the extra treatment time, as well as the double CBCT patient exposure. The authors should point out the need for their suggested ART methodology. Probably, this could be more essential in stereotactic treatments, where the delivered dose is much higher.
Response: We sincerely appreciate your question, which is really important in ART technique. Virtual PTV (+1cm) volumes (304.8cc) are similar to the largest plan library volume of PTV (30-45) (306.cc). However, only 16.6% of the cases received treatment that was delivered using PTV 30-45 plans. In addition, several studies recommended margins that are larger than 1 cm that take into account patient motion, bladder filling, and bladder centroid motion. While it is true that ART therapy requires a lot of extra work and effort, it is worth considering the benefits that ART can provide to patients.
Line 193. Please refer that these 12 post-RT CBCTs belong to the “poor” classification group.
Response: Accordingly, we have revised the manuscript as follows:
“Of the 408 post-treatment CBCTs, 12 (0.03%) were in the “poor” classification group and showed the bladder as partially outside the PTV margin.”
Line 221. In this paragraph, the authors compare the 2-years OS and CSS from their study with the 5-years outcomes from other studies. It would be preferable if the authors compared the outcomes of the same time periods, in order to critically evaluate the results from the Adaptive RT.
Response: Accordingly, we have revised the manuscript as follows:
“The OS and CSS rates at 5 years were 36.4% and 66.7%, respectively. The non-muscle invasive local recurrence survival rates at 1 and 2 years were 100% and 66.7%, respectively (Figure S2), showing a high local control late. Published long-term outcomes following TMT vary greatly, ranging from 26% to 73% for 5-year OS and 30% to 82% for 5-year CSS. The rate of survival in our study was comparable…”
Additionally, the adverse events (acute and late) from adaptive and conventional RT should be compared.
Response: In our study, we retrospectively analysed 19 patients who underwent RT using individualized plan libraries for MIBC. As all patients underwent adaptive RT, comparing adverse events from adaptive and conventional RT is not feasible.
Line 281. The authors mention dosimetric results from other studies. In my opinion, this study should also present dosimetric results for the organs at risk, compare them with the conventional RT doses, and correlate them with the statement of ‘less RT-related toxicity’.
Response: In our study, the mean volume of the PTV libraries was significantly smaller than that for the non-adaptive conventional PTV volumes (CTV+1 cm and CTV+1.5 cm). By applying our results to these previous dosimetry studies, it can be inferred that a smaller PTV volume reduced the volume of irradiated normal organs. However, compared with non-adaptive PTV, the dosimetry results of normal organs of ART PTV could not be analysed in this study due to time limitations and various requirements. However, in the future, we hope to conduct a planning study. Until then, we can only infer that less RT-related toxicity occurred. We mentioned this in the limitations of this study (Line 293–294) as follows:
“The limitations of this study include selection bias and a lack of comparative analysis as regards irradiated normal tissue.”
Reviewer 2 Report
Your study presents a very careful and practicable method of radiotherapy for the bladder, an organ of variable size. I think it is a good way to extend the indications for intensity-modulated radiotherapy to such a moving organ.
1. Has the dose prescription been done for the D95 of the PTV?
2. It is good that your study shows that the treatment time is short (average 9 minutes, maximum about 12 minutes), is there anything you have done to make it shorter?
3. I think your treatment method is very careful, but it seems to take a lot of time and effort to plan the treatment. (It simply seems to take almost three times longer than usual.) How long did it take from the time the simulation CT was taken to the start of treatment?
4. Would you consider increasing the dose to the lesion to obtain a higher local control rate?
5. In which organs did distant metastases occur after treatment? And does your study suggest that irradiation of the lymph node area is not necessary?
Author Response
We appreciate your attention and helpful comments, which have contributed significantly to this manuscript. Please note that the changes made do not influence the content, conclusions, or framework of the paper. We have not listed below all minor changes made; however, these are highlighted in yellow in the revised manuscript.
Reviewer #2
Your study presents a very careful and practicable method of radiotherapy for the bladder, an organ of variable size. I think it is a good way to extend the indications for intensity-modulated radiotherapy to such a moving organ.
Response: We sincerely appreciate your positive feedback.
- Has the dose prescription been done for the D95 of the PTV?
Response: We prescribed D90 of the PTV.
- It is good that your study shows that the treatment time is short (average 9 minutes, maximum about 12 minutes), is there anything you have done to make it shorter?
Response: Thank you very much for your critical and important question. To shorten the treatment time, the treatment plan was decided using dual arc volumetric modulated arc therapy (VMAT). This is mentioned in “Materials and Methods - 2.4 Treatment planning” (Line 104–105) as follows:
“To minimize bladder filling during treatment, it was necessary to shorten the treatment time. Therefore, the treatment plan was decided using dual arc volumetric modulated arc therapy (VMAT).”
- I think your treatment method is very careful, but it seems to take a lot of time and effort to plan the treatment. (It simply seems to take almost three times longer than usual.) How long did it take from the time the simulation CT was taken to the start of treatment?
Response: Thank you very much for pointing out the time and effort taken to plan the treatment. With the help of several colleagues in our department, treatment could be initiated within 5 days after CT simulation.
- Would you consider increasing the dose to the lesion to obtain a higher local control rate?
Response: The LRFS rate was as high as 87.5% at 1 year in our study. We look forward to the results of the Phase II RAIDER trial (NCT02447549) that will randomize MIBC patients to either standard whole-bladder RT, standard-dose ART, or dose-escalated ART; the results of this trial are predicted to shed more light on the clinical benefits of ART. If this study proves that increasing the dose is helpful for obtaining a high local control rate, we have a plan to actively implement those schemes.
- In which organs did distant metastases occur after treatment? And does your study suggest that irradiation of the lymph node area is not necessary?
Response: Thank you for your question, which is important in clinical practice. There were three patients that experienced distant metastasis. One patient experienced recurrent bladder cancer with pelvic LN and bone metastasis. One patient experienced brain metastasis, and another patient experienced lung metastasis.
The irradiation dose and field vary greatly depending on each guideline and remain unstandardized. In the BC2001 trial, clinical lymph node-negative patients were included and the irradiated target was the bladder. Pelvic lymph node failure was reported to be only 4.9% after CCRT. In the KROG 14-16 study, pelvic lymph node failure was reported to be 1.2% (one patient) among the 85 patients after CCRT. Similarly, in our study of eight cases that experienced local and distant metastasis, only one case developed lymph node metastasis. These results imply that RT fields, including bladder only, could be feasible in selected cT2-4N0 MIBC and that concurrent chemotherapy might also target microscopic lymph nodal disease.